# Applying a Health Equity Lens to Work-Related Motor Vehicle Safety in the United States

**DOI:** 10.3390/ijerph20206909

**Published:** 2023-10-11

**Authors:** Stephanie Pratt, Kyla Hagan-Haynes

**Affiliations:** 1National Institute for Occupational Safety and Health, Division of Safety Research, Morgantown, WV 26505, USA; sgp2@cdc.gov; 2Strategic Innovative Solutions, LLC, Clearwater, FL 33760, USA; 3Injury and Violence Prevention Center, Colorado School of Public Health, University of Colorado Anschutz Medical Campus, Aurora, CO 80045, USA; 4National Institute for Occupational Safety and Health, Western States Division, Denver, CO 80225, USA

**Keywords:** equity, occupational road safety, crash disparities, accident prevention

## Abstract

Motor vehicle crashes (MVCs) are the leading cause of fatal work-related injuries in the United States. Research assessing sociodemographic risk disparities for work-related MVCs is limited, yet structural and systemic inequities at work and during commutes likely contribute to disproportionate MVC risk. This paper summarizes the literature on risk disparities for work-related MVCs by sociodemographic and employment characteristics and identifies worker populations that have been largely excluded from previous research. The social–ecological model is used as a framework to identify potential causes of disparities at five levels—individual, interpersonal, organizational, community, and public policy. Expanded data collection and analyses of work-related MVCs are needed to understand and reduce disparities for pedestrian workers, workers from historically marginalized communities, workers with overlapping vulnerabilities, and workers not adequately covered by employer policies and safety regulations. In addition, there is a need for more data on commuting-related MVCs in the United States. Inadequate access to transportation, which disproportionately affects marginalized populations, may make travel to and from work less safe and limit individuals’ access to employment. Identifying and remedying inequities in work-related MVCs, whether during the day or while commuting, will require the efforts of industry and multiple public sectors, including public health, transportation, and labor.

## 1. Introduction

In the United States (U.S.), longstanding policies at all levels of government, the actions of the private sector, and social norms and attitudes have led to intersecting systemic barriers to success for persons who belong to marginalized groups. Because of their social, demographic, employment, or economic characteristics, persons in marginalized groups cannot access the full benefits that accrue to other members of society [1,2,3,4]. Not surprisingly, these systemic barriers have also led to significant health and safety inequities. While many discriminatory or exclusionary policies are no longer explicit, systemic racism and other forms of oppression persist, as do the health and safety inequities associated with them. One significant area of inequity is related to disparities in the risk of motor vehicle crashes (MVCs), which is closely associated with access to safe and adequate transportation [5,6]. Achieving transport justice, which is a component of social justice, requires a fair distribution of risks and benefits, which means that planning and executing transport policies need to go beyond purely technical considerations [7]. If transport justice is to be achieved, local residents should have a voice in the transportation policy decisions that will affect their safety and well-being [8].

The U.S. government’s commitment to identifying and eliminating economic, social, and health inequities is demonstrated by an Executive Order directing each federal agency to “assess whether, and to what extent, its programs and policies perpetuate systemic barriers to opportunities and benefits for people of color and other underserved groups” [9]. In response, the U.S. Department of Transportation has implemented an Equity Action Plan, designed to address inequities and ensure that social justice and benefits to historically marginalized groups are embedded in policy decisions [10].

In the U.S., MVCs are the third leading cause of death from unintentional injuries among persons of all ages, after poisoning and falls [11], accounting for 38,824 deaths in 2020 [12]. However, the risk of MVCs, along with resulting injuries and fatalities, is not evenly distributed across demographic, economic, and social groups. Data from the Fatality Analysis Reporting System (FARS), which is the national census of police reports on fatal road traffic crashes in the U.S., indicate that males have 2.7 times the MVC fatality rate of females, and that fatality rates are highest for persons 21–24 years of age, declining gradually before increasing again at age 75 [12]. Among racial groups, people who identify as American Indian or Alaska Native and Black have the highest MVC fatality rates [13].

For millions of U.S. workers, driving a motor vehicle or being exposed to traffic hazards as a pedestrian while at work is a source of injury or fatality risk. MVCs are the leading cause of work-related fatalities in the U.S., accounting for 35% of all deaths in 2019 [14]. As with MVCs in the general population, there are disparities in the risk of fatal work-related MVCs by sex, age, and race/ethnicity. Frequencies and rates of crashes and fatalities are generally reported to be higher for males than for females [15,16,17,18], and older workers have the highest fatality rates [16,19]. American Indian and Alaska Native workers and Black workers have the highest rates of fatal work-related crashes [15,16].

Disparities in MVC risk do not originate from biological, genetic, or physiological characteristics [2,20]. Rather, they are largely attributable to differences in social determinants of health (SDOH), which arise from differences in the access and quality of healthcare and education, economic stability, the social and community environment, and the neighborhood and built environment [21]. The contribution of work factors to MVC disparities is absent from literature that addresses the relationships between SDOH and MVCs. To assess the role of work in MVC disparities, researchers and policymakers should consider the broad social, political, and organizational environments that foster and perpetuate these disparities.

In this commentary, we review international literature on the current state of knowledge on disparities in work-related MVCs. We use the social–ecological model (SEM) to identify potential sources of risk disparities, in primarily the U.S. context, at five levels—individual, interpersonal, organizational, community, and public policy. We then identify worker populations that have been excluded from research and discuss data collection and research that is needed to understand and address MVC disparities.

## 2. Disparities

### 2.1. Work-Related Motor Vehicle Crashes

#### 2.1.1. Sex

Overall, males are over-represented among work-related MVCs and have higher injury and fatality rates than females; rates of crashes or fatalities for males are often 6–8 times those for females [15,16,17,18,22]. However, some studies specific to light-vehicle use have reported higher crash, injury, or fatality risk for female drivers [23,24,25]. Further, U.S. studies of crashes involving large trucks found that females were more likely than males to suffer a fatal or incapacitating injury [26] or a serious injury [27].

#### 2.1.2. Age

Studies that calculated work-related MVC rates or assessed relative risk by age reported mixed results. Several studies found that although the highest proportions of injuries and fatalities generally occurred among middle-aged groups, it was older workers who were at the highest risk [15,16,18,19,22,28,29]. In contrast, the highest fatality rates in the U.S. mining sector were reported to be among workers ages 16 to 19 years and the lowest rates at age 65 or older [30].

Other research has shown that older workers are more likely to be injured if they are involved in a crash and more likely to die if they are injured [19,26,31,32,33]. In contrast, Newnam et al. (2018) did not find that crash-involved workers aged 60 or older were statistically more likely to be injured or killed compared to younger workers [34].

Studies of nonfatal work-related MVCs did not find that the oldest workers were at the highest risk of crashes or injuries. A U.S. study of persons treated in hospital emergency departments reported the highest rates for workers ages 15–19, with slightly lower rates for workers age 70 or older [35]. Similarly, an Australian study of truck drivers reported significantly higher rates of collisions per million miles (CPMM) among drivers age 25 or younger, with lower CPMMs among drivers age 65 or older [36]. Another study that used vehicle miles traveled (VMT) as the rate denominator found that drivers aged 65 and older had significantly lower CPMM rates than younger drivers [24].

#### 2.1.3. Race and Ethnicity

The literature reports data on work-related MVCs by race and ethnicity in different ways, and results are not consistent across studies. Two studies reported the highest risks for American Indian and Alaska Native workers and Black workers [15,16]. A New Zealand study reported that MVC fatality rates for Māori workers were nearly three times that of other ethnic groups [22]. However, many papers that reported data by race contained few categories, such as White and Black workers only, White and non-White workers, or workers who were White, Black, and of other races [29,37,38]. Another common practice was to treat race and ethnicity as a single construct, combining them so that the values summed to the total number of cases [15,39,40]. One analysis provided rates by race and ethnicity but did not cross-tabulate the two variables [16]. Other papers provided percentage distributions but no rates [39,41]. One paper assessed the relative risk of a fatal work-related MVC by Hispanic ethnicity, finding no significant differences, but did not examine disparities by race [19].

### 2.2. Vehicle Ownership and Reliability

The availability of a vehicle can substantially affect one’s quality of life and opportunities for employment and is closely linked to income. An estimated 8.6% of U.S. households in 2010 did not have access to a vehicle, with higher proportions among households headed by persons of color (19.8% with a Black or African American householder, 13.3% with an American Indian or Alaska Native householder, and 12.7% with a Hispanic householder) [42]. Another source indicates that in 2017, 24.3% of lower-income households had no access to a vehicle [43].

The lack of a vehicle limits the type of job an individual can obtain as well as the ability to access that job. This might be a greater impediment in car-dominated societies, such as the U.S., or in locations where public transportation is unavailable or modes of transport are not linked together. It might also result in longer and more burdensome commutes. Transit delays that lead to frequent tardiness can affect a worker’s ability to keep a job. Employers might even include questions about access to a vehicle or reliable transportation on job applications, potentially closing off employment opportunities for some [44]. In low-income households that do have a vehicle available, vehicles are older and less reliable. The mean vehicle age in 2017 for households earning 15,000–24,999 U.S. dollars (USD) annually was 12.48 years, compared to 8.84 years for households earning 100,000–124,999 USD annually [45]. Older vehicles also lack the advanced safety features found on newer vehicles [46].

### 2.3. Multiple Sources of Disparity and Overlapping Vulnerabilities

A major limitation of looking separately at demographic and employment variables such as sex, age, race, ethnicity, occupation, and industry is that it does not properly account for an individual identifying as a member of multiple marginalized groups. The concept of intersectionality explains that if we analyze single sources of disparity independently, we obscure the inequities that accrue from being part of multiple disadvantaged or marginalized groups [47]. In proposing intersectionality, Crenshaw used the example of Black women to show that efforts to achieve gender equity have privileged White women, and efforts to achieve racial equity have privileged Black men; thus, Black women are doubly disadvantaged. It follows that the concept of intersectionality extends to inequities in the workplace. In the work setting, this has been referred to as “overlapping vulnerabilities” and “cumulative precariousness.” One source used “overlapping vulnerabilities” to characterize how interactions between multiple risk factors might result in greater risk than would a single risk factor [48]. Similarly, another source used “cumulative precariousness” to describe how unstable employment arrangements, economic disadvantage, and immigrant status combine to create conditions under which workers with more than one of these characteristics experience disproportionately poor working conditions and health outcomes [49].

In many occupations with high exposure to driving and traffic, workers who belong to communities of color make up a disproportionate share of the U.S. workforce. In addition, non-Hispanic Black workers and Hispanic workers in these occupations have lower median incomes than non-Hispanic White workers in the same occupations (Table 1) [50]. Only among parking attendants and bus, shuttle, and taxi drivers did non-Hispanic Black workers or Hispanic workers have higher median incomes than non-Hispanic White workers.

## 3. Applying the Social–Ecological Model (SEM) to Understand Potential Factors Underlying Work-Related Crash Disparities

Recent scholarship supports a shift from a biomedical model of injury prevention, which is based on preventing the harmful exchange of energy, to a biopsychosocial model, which considers the social context in which injuries occur [51,52]. In this paper, we expand the framework suggested by Flynn et al. (2022) [51], using the SEM proposed by McLeroy et al. (1988) [53] to more fully delineate factors that might contribute to disparities in work-related MVCs.

The SEM places the individual at the center (Figure 1). Individual factors include biological or intrapersonal factors that contribute to MVC crash risk or crash-related injury. Individual factors must also encompass off-work risk factors, which are sometimes considered outside the OSH domain yet have been shown to be associated with the risk of work-related crashes, such as inadequate sleep and use of medications and substances [54,55,56].

In the SEM, individual factors are embedded within progressively larger circles depicting interpersonal relationships, organizational factors, community-level factors, and public policy. This conceptualization is clearly relevant to disparities in the risk of work-related MVCs, as it illustrates the breadth and interaction of factors outside an individual’s control that can create these disparities. Organizational factors are especially relevant to work-related MVCs, as they include the employer’s culture and policies, employment arrangements, and industry-specific norms and practices that might inhibit MVC prevention efforts and influence individual behaviors and decisions.

Emphasis on the four layers outside the individual is consistent with the general paradigm shift away from a focus on individual behavior to the social context, as has occurred in traffic safety in recent decades. The Safe System approach, advocated for by Vision Zero networks dedicated to the elimination of road fatalities globally, shifts much of the responsibility for road safety to the parties whose actions can eliminate the unsafe conditions that lead to road users’ errors—road designers, vehicle designers, policymakers, and first responders [57,58,59]. For work-related MVS, employers are integral to a Safe System. The international road safety management standard recommends the use of risk management principles by employers [60]. This approach is consistent with the Safe System.

In this section, the SEM framework is applied to equity research on work-related MVCs. This framework can guide researchers in identifying and suggesting remedies for sources of inequity that originate at all levels of the SEM.

### 3.1. Individual Factors

#### 3.1.1. Sex

The difference in serious injury rates by sex may be due in part to differences in the makeup and structure of bones and ligaments [61,62]. Further, females wearing seat belts are more likely than male drivers wearing seat belts to be injured when involved in a comparable crash [63]. These disparities have been attributed to sex differences in the bone structure and musculature of the neck, the placement of head restraint devices, females’ relatively short stature, and females’ preferred seating posture [63].

#### 3.1.2. Age

Research results showing higher crash severity and mortality for older persons involved in work-related MVCs are consistent with data for the general population [46]. Greater physical frailty and likelihood of co-morbidities are a large part of the explanation for older persons’ higher risk of injury or death [64]. The increased risk for older persons also extends to pedestrians; a 70-year-old struck by a vehicle traveling at 25 mph is as likely to die or be severely injured as a 30-year-old struck at 35 mph [65].

The safety of vehicles may also contribute. Older persons are more likely to drive older vehicles and vehicles that are less likely to have advanced safety features [46], which might affect both crash incidence and survivability. This might well apply to older workers who drive their own vehicles for work.

### 3.2. Interpersonal Factors

The quality of the relationship between management and workers has driver safety implications, as these relationships are an important element in drivers’ motivation to drive safely [66]. Frequent safety-related discussions between supervisors with their workers have been shown to be associated with safer driving behaviors [67]. In addition, drivers reported higher motivation to drive safely when they perceived that their supervisors and fleet managers valued safety [66].

The interactions of occupational drivers and pedestrian workers with their managers, co-workers, and customers are likely different for those who belong to marginalized groups than for their White peers, but little research has addressed these differences. More broadly, workplace discrimination against persons who belong to communities of color, the LGBTQ+ community, and other marginalized groups is well-documented, as are the impacts of workplace discrimination on employees’ physical and mental health. For example, Fekedulegn (2019) found that 25% of Black women aged 48 years or older reported workplace discrimination as compared to 11% of White men [68]. In addition, individual sources of job stress, such as workplace bullying, have also been shown to be disproportionally experienced by women and people of color [69].

Interpersonal workplace conflicts and job stress in general are associated with risky driving behaviors for occupational drivers [70,71,72]. A study of bus rapid transport drivers in Colombia found that stressful working conditions predicted risky driving behaviors and that a lack of social support and job strain were associated with fatigue, which in turn contributed to risky driving behaviors [73]. The stressors of professional driving are also associated with negative physical [74] and mental health outcomes [72]. None of these studies reported differences in job stress or risky driving by race, ethnicity, or other social factors. More generally, however, compared to White, non-Hispanic workers, Black workers in the United States were found to be over-represented in occupations with negative psychosocial working environments that offered less autonomy, opportunity for advancement, or recognition [75].

Language and cultural barriers between managers and workers might impede effective risk communication. For example, the authority to “stop work” in circumstances where safety is at risk, which is based on the assumption that workers are aware of what is safe and unsafe, is common in high-risk industries [76]. Members of marginalized worker populations, such as immigrant workers who do not have the documentation needed to prove legal residency (often referred to as “undocumented immigrants”), may be less inclined to speak up or “stop work” [77,78]. A “need to justify stop work decision to management” and “being afraid of others” were among many factors cited by petroleum workers as barriers to their ability to stop work [76]. Lower-paid workers may also feel pressured to continue driving despite unsafe conditions such as fatigue because they need the money; petroleum workers cited the “need to complete the job in order to receive payment” as an additional barrier to stopping work [76]. Drivers may also fear retribution for stopping work due to unsafe conditions if they have experienced discrimination in other circumstances.

Language and cultural barriers also affect the likelihood that immigrant workers and low-literacy workers will receive safety training, including driver training, which is appropriate for their language fluency and literacy level. Limited English proficiency might mean that workers whose employers do not provide multilingual training will not fully understand job tasks and safety procedures [77,79].

Workplace programs that aim to improve communication and support the inclusion of marginalized workers (for example, training in diversity, equity, and inclusion (DEI) and cultural humility) are increasingly common across industries. However, limited evaluation of the success of such programs has been completed, and no information is available on the effects of programs on general safety or MVS outcomes.

### 3.3. Organizational Factors

The organizational circle of the SEM encompasses a number of factors that are associated with the risk of work-related MVCs, including employer policies, employment arrangements and work schedules, compensation and payment methods, and remote work and telework policies. However, very little research has assessed the relationships between these factors and other sources of inequity, such as race and ethnicity, economic status, age, and sex.

#### 3.3.1. Employer Policies

Employer MVS policies, such as mobile phone record checking following crashes, fatigue management, driver training, and collision response procedures, have all been associated with lower collision rates [80]. The extent to which these policies are implemented among companies within the U.S. and the extent to which drivers who belong to marginalized groups are protected by them is unknown.

A culture of organizational learning where employees are encouraged to view mistakes as learning opportunities is also linked to lower crash rates for organizations employing professional drivers [81]. Given that some components of MVS policies can be punitive instead of positively reinforcing, questions arise as to the equitable application of such policies, including practices such as phone record checking following a crash and the follow-up investigation of crashes. The distribution of supportive versus retributive MVS policies along racial and ethnic lines may warrant further exploration.

#### 3.3.2. Employer Arrangements and Work Schedules

Employment arrangements are linked to safety outcomes through disproportionate exposure to work-related employment benefits and safety and health risks [82]. Workers with work arrangements defined as non-traditional or alternative include independent contractors, on-call workers, temporary help agency workers, and workers provided by contract firms. Differences in safety culture, training, availability of personal protective equipment, and assignment of dangerous tasks may lead to worse safety outcomes for non-traditional workers [83]. The May 2017 Contingent Worker Supplement to the Current Population Survey estimated the proportion of U.S. workers considered to have alternative work arrangements: independent contractors (6.9%), on-call workers (1.7%), temporary help agency workers (0.9%), and workers provided by contract firms (0.6%). Hispanic workers are more likely to report alternative employment arrangements than non-Hispanic White workers [84,85]. Limited research has estimated differences in occupational crash rates between drivers working in traditional versus alternative arrangements. However, an analysis of MVC fatalities among oil and gas extraction workers revealed significantly higher motor vehicle crash fatality rates for employees of contract firms as compared to other employees [41].

Another relevant employment characteristic is job security. Contingent workers, defined as those who report that their jobs are temporary or who do not expect their jobs to last, are estimated to be 1.3% to 3.8% of the workforce, depending on the model used [86]. Unpredictable work schedules are common for contingent workers, and these schedules are associated with psychological stress and poor sleep quality, risk factors for fatigue-related crashes [87,88]. Hispanic workers are substantially more likely to report job insecurity than non-Hispanic White workers [84,89]. However, the extent to which workers who belong to marginalized groups are affected by MVC risks linked to job insecurity is unknown.

Another aspect of employment arrangements that might be associated with disparities in MVC risk is the multi-contractor structure found in industries such as construction and oil and gas extraction, both of which have a large Hispanic workforce. The crash fatality rates for both industries are elevated as compared to the general working population [41,90]. Multiple layers of contract relationships may be associated with poor communication about OSH and inconsistent and inadequate application of OSH policies at subcontractor levels [91].

Long work hours and extended work shifts are associated with an increased likelihood of falling asleep while driving and being involved in an MVC [90,92]. Drivers involved in sleep-related crashes were more likely to have night jobs, multiple jobs, or another type of non-traditional work schedule [93]. Non-Hispanic Black workers were more likely to report shift work and alternative shift schedules such as night or rotating shifts than non-Hispanic White workers [84,94]. Persons 18–29 years of age were also more likely to work alternative shifts than any other age group. 

#### 3.3.3. Compensation and Payment Methods

Commercial motor vehicle (CMV) drivers are an occupational group that experiences systemic inequities affecting quality of life as well as crash and injury risk. Mean annual income for heavy and tractor-trailer truck drivers in 2020 was 48,710 USD versus 56,310 USD for the average U.S. worker [95]. For truck drivers, earning their desired level of income involves long hours and does not include compensation for overtime. CMV drivers in the U.S. are exempt from the Fair Labor Standards Act, which mandates overtime pay for work beyond 40 h a week [96]. On the other hand, Newnam et al. (2017) found that in Australia, greater compensation of occupational drivers was associated with safer driving behaviors, but only when a positive safety climate was in place [97].

Truck drivers are also often paid by the mile or the load, not by the hour, which can incentivize unsafe driving, and many are not paid at all for time spent waiting, loading, or unloading [98]. Being paid by the load has been shown to be associated with increased sleepiness, fatigue, and crashes among truck drivers [99,100]. In 2010, U.S. long-haul truck drivers reported working an average of 60.4 h per week, 16.2 h of which involved non-driving tasks for which they might not have been paid [101]. Truck drivers are also often paid by the mile or the load, not by the hour, which can incentivize unsafe driving, and many are not paid at all for time spent waiting, loading, or unloading [98]. Being paid by the load has been shown to be associated with increased sleepiness, fatigue, and crashes among truck drivers [95,96]. In 2010, U.S. long-haul truck drivers reported working an average of 60.4 h per week, 16.2 h of which involved non-driving tasks for which they might not have been paid [97].

Occupational drivers who act as independent contractors may also be impacted by payment structures. For example, delivery and rideshare drivers are likely to feel pressure to drive during times of peak payment rates and may not be able to plan for sufficient rest during such periods.

#### 3.3.4. Remote Work/Telework Policies

To reduce exposure to the risk of a crash at work or while commuting, working from home confers obvious benefits. However, the ability to work from home is an advantage that is not equally distributed among racial and ethnic groups, and the COVID-19 pandemic has exacerbated this disparity. First, an individual needs to have a job that allows them to work from home, and second, most jobs that can be done from home require reliable high-speed internet access. A recent analysis showed that in the U.S., Hispanic workers are over-represented in jobs that are not suitable for working from home, and Black workers are over-represented in jobs with higher potential COVID-19 exposure [102]. Although access to broadband internet has increased substantially in recent years among all racial and ethnic groups, broadband access is still lower among Hispanic and Black households. In 2021, 65% of Hispanic households and 71% of Black households had broadband access, compared to 80% of White households [103].

### 3.4. Community Factors

#### 3.4.1. Road Infrastructure

Characteristics of road infrastructure and the road transport system are associated with MVC crash risk among the general population, including four-way intersections, poor lighting/visibility for pedestrians, uncontrolled access points, congested traffic, poorly designed curves, and uncontrolled railroad crossings [104]. However, we are not aware of any research that explores the effects of poor road infrastructure on the safety of occupational drivers, pedestrian workers, or pedestrian commuters.

Differences in the quality of road infrastructure are associated with disparities in MVC risk between higher- and lower-income neighborhoods. Research conducted in Montréal, Canada, found that on average, lower-income neighborhoods had 6.3 times more pedestrian injuries, 3.9 times more cyclist injuries, and 4.3 times more motor vehicle occupant injuries than higher-income neighborhoods [105]. Significant contributors to these disparities included an excess of four-way intersections and major roads, greater population density, and higher traffic volume [105]. In this study, workers who lived in low-income neighborhoods were more likely than those in higher-income neighborhoods to walk, cycle, and use public transport to commute to and from work.

Persons who live and work in rural areas may lack access to public transportation and are likely to depend on a motor vehicle to commute to and from work and between work locations. Travel on rural roads is considerably riskier than on urban roads; in 2020, the MVC fatality rate on rural roads in the U.S. was 1.7 times that for urban roads [106]. Oil and gas extraction work involves frequent travel in remote areas with poor road infrastructure, which might contribute to the elevated MVC fatality rate for this industry [107].

#### 3.4.2. Commuting Opportunities and Behaviors

Commuting opportunities and behaviors are a critical element of the social and economic contexts that place some workers at greater risk of work-related MVCs. MVC commuting disparities may be linked to income, rural residence, employment arrangements, access to a vehicle, access to public transportation, and road infrastructure quality. Research among U.S. oil and gas extraction workers found that mean commuting time had a significant positive association with feeling drowsy and falling asleep while driving a work vehicle and having had a “near miss” crash in the past week [56]. Further, as international data show, commuting is a separate source of crash risk for workers who must use the road to travel to and from work. In Europe and Australia, commuting-related crashes are included in OSH data systems due to their coverage under workers’ compensation (WC). Studies from these regions indicate that commuting crashes are a significant injury and fatality burden, in many cases eclipsing crashes that occur during the workday [32,33,108,109].

Commuting crashes may disproportionately affect marginalized U.S. worker populations, who may experience more serious commuting-related risk factors. Many of these risk factors pertain to commuting on foot and/or via public transport, but some are related to vehicle access and characteristics. For lower-income workers and those who belong to communities of color, commuting time (and, hence, exposure time) is greater due to many factors, including lack of affordable housing near work or living in a household without a car [42,43]. Lower-income workers without access to a car spend more time as pedestrians, walking to and between transit modes, and waiting at transit stops. Workers with non-standard or weekend schedules are further disadvantaged because of reduced transit options at off-peak hours [110]. Travel to and from work during off-peak hours may also increase MVC risk for pedestrians, as 73.6% of pedestrian fatalities in the U.S. in 2020 occurred between the hours of 6 p.m. and 6 a.m. [12].

Residents of neighborhoods with larger concentrations of lower-income persons and persons who belong to communities of color live and walk to work or public transportation in neighborhoods with inadequate road infrastructure, including poor pedestrian facilities [111]. Historically, these neighborhoods have seen under-investment in road infrastructure. Further, until recently, the primary goal of transportation planning has been to build roads that optimize speed and mobility, thereby favoring car drivers over pedestrians and other road users [5,58,111,112]. Ignoring the needs of pedestrians and other road users in the planning of transportation networks and road-building projects has compromised their safety as well as their ability to access employment and services conveniently and affordably.

### 3.5. Public Policy Factors

Workers who belong to marginalized populations are at disproportionate risk of a work- or commuting-related MVC because of inadequate, exclusionary, or discriminatory policy in both the labor and transportation sectors.

#### 3.5.1. U.S. Federal Regulations

In the realm of labor policy, the Federal Motor Carrier Safety Regulations (FMCSRs) represent the only body of U.S. federal OSH regulations related to driving for work, applying to CMV drivers (i.e., drivers of large trucks and buses). The FMCSRs cover operational areas such as hours of service (HOS), vehicle safety standards, drugs and alcohol, commercial driver licensing, training, medical fitness for duty, and mobile phone use (49 CFR Parts 300–399). The extent to which U.S. occupational drivers of different races and ethnicities are covered by the FMCSRs is unknown, as there are no available estimates of persons employed as heavy and tractor-trailer truck drivers by race and ethnicity. However, employment in the truck transportation industry (all types of jobs) in 2021 was skewed toward White persons (74.5% versus 18.7% for Black persons and 23.6% Hispanic persons of all races) [113], so persons in driving jobs might be similarly distributed by race and ethnicity.

Evaluation of the effectiveness of the FMCSRs in improving the safety of CMV drivers is limited. Researchers have assessed the fatigue potential of the 34-h “restart” element (required off-duty period after reaching the maximum weekly driving limit) of the HOS regulations [114] and the potential safety benefits of electronic HOS recorders [115], but these studies did not stratify data by sociodemographic variables that might account for disparities in the adoption or effectiveness of these measures.

Although the FMCSRs offer some protections for CMV drivers, they exempt some workers from the HOS regulations that govern maximum driving and duty hours, creating risks for driver fatigue for these workers. Worker groups for whom HOS relaxations or exemptions are available include utility service drivers, oil well servicing drivers, and drivers transporting construction equipment and agricultural products (49 CFR Part 395). The risk of a fatigue-related MVC is exacerbated by the exemption of CMV drivers from overtime provisions of the Fair Labor Standards Act which, as noted earlier [96], creates incentives to work longer hours to earn more money.

#### 3.5.2. Voluntary Standards and Industry Guidance

In contrast with drivers of large trucks and buses, drivers of light vehicles are not covered by any federal regulations equivalent to the FMCSRs; therefore, managing their MVC risk depends on state traffic laws and employer initiatives. Voluntary consensus standards have been developed to assist employers in managing MVC risk [60,116] and industry and membership organizations have developed their own guidance [117,118,119]. However, the extent to which these standards and guidance have been implemented is unknown. It follows that the extent to which these voluntary recommendations have been applied to workers who belong to marginalized groups is also unknown.

#### 3.5.3. Workers’ Compensation Programs

In the U.S., WC programs are administered by the states, which determine their own parameters for coverage and compensation. National estimates indicate that in 2020, about 136 million jobs were covered by WC, but these estimates were not stratified by race, ethnicity, age, occupation, or other indicators [120]. State WC programs are generally likely to exclude the self-employed, gig workers, agricultural and domestic workers, and independent contractors [120,121]. A study of work-related injuries treated in hospital emergency departments found that 60.4% of workers expected WC to be the primary payer, but that Black workers of all ethnicities were significantly less likely than White workers of all ethnicities to cite WC as the primary payer [122]. A separate analysis of expected primary payers by the worker’s ethnicity found no significant differences between Hispanic and non-Hispanic workers. Lower levels of self-reported WC coverage among Black workers might be attributable to self-employment, greater employment in jobs with contingent or alternative work arrangements, or employment in low-wage jobs where employers can opt out of providing WC coverage [122]. In addition, Black workers might have been missing from the data set altogether due to reluctance to report a work-related injury.

#### 3.5.4. Policies That Affect Immigrants

For undocumented immigrant workers in the U.S., state driver licensing laws have implications for employment and safety. Only 18 U.S. states and the District of Columbia currently allow undocumented immigrants to obtain a driver’s license [123]. These exclusions close off employment opportunities, including the opportunity to obtain a driving-related job. They also close off options to commute to and from work using a vehicle. Workers who choose to drive without a license would be less likely to understand traffic laws and would not be able to obtain insurance, placing them at financial and legal risk.

The consequences of being injured in an MVC at work or while commuting might be more severe for immigrants than for other workers, particularly if they are not documented. A review of the literature on access to healthcare identified a range of barriers at individual, institutional, and policy levels [124]. If injured in a work-related MVC, immigrants might have difficulty obtaining and paying for medical care; in 2020, among the nonelderly population, 26% of lawfully present immigrants and 42% of undocumented immigrants did not have health insurance, compared to 8% of U.S. citizens [125]. Low levels of health insurance coverage for immigrants in the U.S. have been linked to ineligibility for public health insurance options as well as immigrants’ over-representation in low-income occupations where employers are unlikely to provide health insurance [126].

Immigrants injured in an MVC might encounter cultural and language barriers to communicating with health professionals, successfully advocating for their interests, and navigating the healthcare system and other bureaucracies. They might also fear being discriminated against or being deported due to their immigration status [124]. In addition, injured immigrant workers might return to work before they are medically ready because their jobs do not provide sick leave or because they need to earn money.

#### 3.5.5. Transportation Policy Decisions

More broadly, transport planning practices have implications for MVS at work and during commuting. Historically, these policies have disregarded the accessibility and mobility needs of low-income persons, persons who belong to communities of color, and persons with disabilities [5,58,111,112]. Low-income neighborhoods are more likely than higher-income neighborhoods to lack pedestrian facilities, making commuting less safe for workers who use public transportation. In addition, road projects have cut through and split up neighborhoods, separating low-income workers from employment opportunities.

## 4. A Call for More Equitable Work-Related MVS Surveillance and Research

### 4.1. Surveillance

#### 4.1.1. Improve and Expand Data Collection

In the U.S., some data items that are critical for advancing our understanding of work-related MVCs are not collected, and existing data are not being used to their full potential. These are significant impediments to research on the systemic and structural causes of disparities in work-related MVCs. While these limitations affect all research on all worker populations, they are likely to have disproportionate effects on research on marginalized populations. In addition, research methods that have been used to examine crashes in the general population have not been applied to work-related MVCs. The following actions are the first steps toward positioning researchers to identify and suggest remedies for inequities in work-related MVS.

#### 4.1.2. Expand Data Linkage and Data Sharing

Data linkage and data sharing have been underutilized for the study of work-related MVCs [127]. There are relatively few examples in the literature [24,108,128,129,130,131,132]. Future research can link data sources, such as the Census of Fatal Occupational Injuries (CFOI) (a multiple-source data system maintained by the U.S. Bureau of Labor Statistics [BLS]), FARS, WC data, company crash and claims data, hospital and emergency department data such as the U.S. National Electronic Injury Surveillance System (NEISS), police crash reports, and motor vehicle records, and conduct analyses by race, ethnicity, socioeconomic status, and other potential sources of disparity.

#### 4.1.3. Incorporate Income and Other Socioeconomic Data in Crash and OSH Data Systems

CFOI includes data items that can help researchers explore potential sources of inequity that might affect fatality risk, including occupation, industry, employment arrangements, and country of birth [133]. However, the CFOI is missing other relevant indicators, notably income. FARS, which is based on police crash reports, does not collect data on economic and social characteristics that might explain why certain populations have higher fatality risk.

The absence of income and socioeconomic data from police crash reports and OSH data systems impedes the identification of disparities by income and other variables associated with income that might increase crash risk, including race, ethnicity, and occupation. Options for incorporating income and socioeconomic data include using occupation as a proxy for income or assigning aggregate income estimates for a small geographic area such as a census tract or block group to individual workers based on their place of residence.

Research data sets should also capture or link to information about medical costs, including costs borne by the worker, workers’ compensation payments, and indemnity payments. Vehicle damage costs and the payer for such costs would also be an important data point with equity implications.

#### 4.1.4. Collect Exposure-Based Denominator Data

Most research on work-related MVCs uses the number of employees or full-time equivalents (FTEs) as the measure of exposure. The primary shortcoming of this approach is that it does not account for differences in driving exposure. Because members of marginalized populations (notably lower-income groups and communities of color) are over-represented in many occupations that involve substantial driving or time spent as a pedestrian, the use of exposure-based denominators will more accurately reflect their risk.

VMT is the exposure measure most widely used by the traffic safety community, but it is not ideal for all situations. Hours of driving have been proposed as a potentially more accurate measure of risk [134]. Some employers are beginning to use this as an alternative to VMT, particularly to assess risks associated with driving in urban areas, which generally involves lower mileage for each hour spent on the road. Mileage-based denominators also do not account for workers’ exposure to crash risk if their job requires them to be a pedestrian or work outside their vehicle.

VMT-based denominators may be biased against persons who do not use vehicles for work and commuting, which might disproportionately affect lower-income persons and persons who belong to communities of color. One study calculated general population rates per 100,000 persons, per 100 million VMT, and per 100 million person-miles traveled (PMT) [135]. Rates by PMT might be relevant for field studies of pedestrian workers and for analyses that assess the risks of commuting on foot.

Not all sources of employment-based denominator data provide estimates needed to calculate rates by demographic characteristics. For example, the Quarterly Census of Employment and Wages (QCEW) is the preferred source of denominator data for the oil and gas extraction industry because it offers separate estimates for operators, drilling contractors, and servicing contractors. However, the QCEW does not provide estimates by age, sex, or race/ethnicity.

#### 4.1.5. Identify Commuting-Related Crashes in Crash Reports and National Data Systems

In the U.S., commuting crashes are excluded from research because no data are available. The BLS definition of a work-related fatality largely excludes incidents that occur while commuting to and from work, thus these incidents are excluded from CFOI [133]. Further, FARS does not ascertain whether a crash involved commuting. The absence of purpose-of-journey data in police crash reports and occupational injury data makes it impossible to quantify the relative burdens of work-related and commuting crashes. The current lack of available commuting data may present an example of structural invisibility, where a lack of data means that inequities experienced by marginalized groups cannot be discovered or remedied [51]. Ascertainment of commuting status in police crash reports and data systems such as FARS, as well as collection of detailed commuting information in research studies based on worker surveys and administrative data, would allow researchers to identify potential inequities associated with the journey to work. Information on the burden and nature of commuting-related crashes in the U.S. would also be useful to transportation planners.

#### 4.1.6. Collect Data on Marginalized U.S. Worker Populations

Neither FARS nor CFOI identifies persons with disabilities or LGBTQ+ persons, thus no data are available to assess the risk and experience of work-related MVCs for people who belong to these groups. Crash disparities and potential sources of inequity for persons with disabilities and LGBTQ+ persons should be explored through the collection of qualitative and quantitative data.

### 4.2. Research

#### 4.2.1. Expand Analyses by Sex

Few studies of at-work crashes focus on worker populations where females make up a substantial portion of the workforce. However, these studies are among the few reporting higher crash, injury, or fatality risk for female drivers [23,24,25]. Of note, Pratt and Bell (2019) is one of the few studies that calculated crash rates based on VMT [24]. Moreover, females in this study drove significantly fewer miles annually than male drivers, supporting the idea that employment-based denominators are not accurate indicators of driving exposure.

In the general population, crash-involved females are at greater risk of serious injury than males, even after age, height, Delta-v (crash forces), body mass index (BMI), and vehicle model year are controlled for [136]. Car safety features are designed for the male body and male seating preferences, and vehicle crash tests use a 50th percentile male dummy, thus vehicle designers currently have inadequate data to help them make cars safer for females [137]. Vehicle safety research should include women in crash and injury severity studies and incorporate female anthropometrics and racial/ethnic data as well. Epidemiological crash research should include occupations and industries with large enough female workforces to make it possible to detect statistical differences by sex.

#### 4.2.2. Expand Analyses by Age

Worker age often appears in occupational research papers in descriptive tables only, with little further discussion, or it is treated as a potential confounder to be controlled for. In addition, research results illustrate the potential drawbacks of using employment-based rates (especially those that are based on total employed persons rather than FTE workers) to assess MVC risk for older workers [24,36]. It is possible that workers progress into less driving-intensive supervisory or managerial jobs as they grow older, self-select into non-driving jobs, or choose to work fewer hours. These differences by age are masked by employment-based denominators, which assume uniform exposure levels.

The mixed results on MVC risk and severity for older persons suggest the need for additional research that considers differences in work hours and the nature of driving by older workers. The use of employment-based denominators might have a disproportionate effect on the validity of research results for older workers, as described above. Ideally, research on older persons who drive for work would use VMT or hours of driving as rate denominators. Calculating FTE rates by occupation and age using data that are already available from the CFOI or the Survey of Occupational Injury and Illness would be a first step. Other analyses of work-related MVCs among older workers might center on pedestrian incidents, which are associated with an increased likelihood of injury and fatality for this population.

In addition, research papers do not report industry-specific injury or fatality rates that are standardized to account for the age distribution of the workforce. For example, without age-standardized rates, it is difficult to compare risk for workers employed in industries such as mining and construction (which tend to have younger workforces) with the risk for workers across all industries. Age-standardized rates would facilitate such comparisons.

#### 4.2.3. Identify the Reasons for Racial and Ethnic Disparities

Few papers report data on work-related MVCs by race and/or ethnicity. The cause for this might be a lack of data or researchers’ decisions to omit these analyses. Whatever the causes, this is an example of structural invisibility, where a lack of data means that inequities experienced by marginalized groups cannot be discovered or remedied [51].

Structural invisibility is reflected by reporting racial/ethnic data in descriptive tables only or treating race and ethnicity as potential confounders. Some papers treated race and ethnicity as a single construct, combining them in a single table section such that the values summed to the total number of cases [15,39]. Most of the research on work-related MVCs with data by race and/or ethnicity is based on U.S. data, the exception being two papers from New Zealand that reported distributions and rates for Indigenous Māori workers [17,22]. In papers from the U.S., race and ethnicity data are handled and reported in so many different ways that it is difficult to draw conclusions across studies. It is also difficult to compare the risk of a fatal work-related MVC with the risk in the general population, as most studies of work-related MVCs lack the necessary detail on race and/or ethnicity.

Hispanic workers are disproportionately employed in jobs where they are on foot all or most of the time, thus they are disproportionately affected by the exclusion of pedestrian workers from research. Hispanic persons made up 17.3% of the U.S. workforce in 2018 but were over-represented in several occupations that involve substantial time working as a pedestrian and being exposed to vehicle traffic: landscaping and groundskeeping workers (51.3%), construction laborers (47.1%), parking attendants (36.2%), and refuse and recyclable materials collectors (32.2%) (Table 1) [138].

For Black workers, the effects of being excluded from MVC analyses are broader. Black workers made up 11.4% of the U.S. workforce in 2018. Like Hispanic workers, they were over-represented in occupations involving work as a pedestrian: security guards (30.8%), parking attendants (28.5%), and refuse and recyclable material collectors (19.3%) (Table 1) [138]. However, Black workers were also over-represented in occupations that involve operating a motor vehicle: transit and intercity bus drivers (39.8%), school bus drivers (30.3%), taxi drivers (26.4%), and shuttle drivers and chauffeurs (25.3%) [138]. Hispanic workers were also over-represented among taxi drivers, shuttle drivers, and chauffeurs, but to a lesser extent than Black workers [138].

Crucially, most research that reports racial and ethnic disparities in the risk of work-related MVCs does not explore the underlying causes of these disparities and how racism might contribute to them. Doing so will require stratification by race and placing race at the center of analysis [2,20]. The literature also emphasizes the importance of understanding the variables that are appropriate indicators for the social and economic contexts in which disparities are found [139]. For work-related MVCs, possible explanatory variables include income, occupation, employment arrangements, vehicle access and quality, work schedule, commuting opportunities, and place of residence. If these data are not available at the person level, data from the U.S. Census Bureau and other agencies, which are available at detailed geographic levels, can be used as proxies.

Consistent with the SEM, it has been suggested that multiple levels of measurement are preferable [139], which would enable multi-level modeling to explain factors within different circles of the SEM. The value of using multiple data sources and mixed methods [20] is illustrated by research comparing self-reports and administrative data, which found evidence of under-reporting of Black workers’ injuries in administrative data [140]. Factor analyses, using an index of variables or a latent construct, and structural equation modeling are appropriate methods for assessing the multilevel and multidimensional role of racism [20]. These methods would all benefit research on racial and ethnic inequities in work-related MVS because racism might be present at several levels. It can be manifested at the community and public policy levels within transportation and housing policy communities that allow inequities faced by marginalized groups to persist [20]. Racism can also be present at the organizational level, where differential treatment of workers of color might place them at disproportionate risk of injury, and at the Interpersonal level, where supervisory practices or attitudes might lead to unsafe working conditions.

To our knowledge, researchers have not explored the possibility that data on nonfatal work-related MVCs are incomplete because under-reporting might be more common among persons who belong to communities of color. Fear of interaction with the police or other authorities might make Black persons reluctant to report a work- or commuting-related MVC. The same might be true for immigrant workers who might be undocumented. Research provides evidence of racial disparities in the reporting of occupational injuries, revealing discrepancies between self-reports and administrative data [140]. Based on self-reports, Black healthcare workers had significantly higher odds of occupational injury compared to White, non-Hispanic workers (OR = 1.93) after adjustment for age, gender, country of birth, self-reported financial distress, and job title. In contrast, based on administrative data, the odds of occupational injury for these same Black workers were not significantly different from White workers (OR = 1.27). Data on MVCs in the general population showed that African-American pedestrians treated at a San Francisco hospital were less likely than White pedestrians (OR = 0.55, *p* ≤ 0.01) to have had a police collision report [141]. This discrepancy might mean that individuals were less likely to report the incident to the police, people at the scene did not notify the police, or police responded but did not file a report. These alternative explanations, which might be related to actions and attitudes at individual, interpersonal, organizational, and community levels, illustrate the applicability of the SEM.

#### 4.2.4. Include Commuting Crashes in Research on Work-Related MVCs

Including commuting crashes in work-related MVC research would allow researchers to identify relationships between MVCs that occur at work and while commuting and associated risk disparities that might stem from inequities. Several studies have assessed crash risk or declines in driving performance during commuting after extended work shifts or night shifts [92,142,143,144,145]. The limited research that assesses both at-work and commuting factors has shown that length of commute can affect the risk of both commuting crashes and at-work crashes; for example, a long commute might lead to fatigue which in turn increases MVC risk at work. On the other hand, shiftwork or long work hours might cause a worker to feel fatigued while driving home from work [56]. Considerable additional research is needed to: (1) further explore how the risk of at-work and commuting crashes are related; and (2) to incorporate variables that might reveal risk disparities, including race, income, and place of residence.

#### 4.2.5. Incorporate Pedestrian Workers into Analyses

Very few studies focus on work-related MVCs among workers in occupations with high exposure to traffic as pedestrian workers, such as those who work as highway maintenance workers, delivery drivers, parking attendants, or refuse collectors. Moreover, studies that report MVC data for all workers do not provide separate tabulations for vehicle occupants and pedestrians. Researchers might not focus on pedestrian workers because compared to vehicle occupants, they account for a low proportion of at-work MVC deaths in the U.S. (18% between 2011 and 2020) [14]. As noted earlier, the exclusion of pedestrian workers or lack of separate tabulations for pedestrian workers excludes workers in certain racial, ethnic, occupational, and socioeconomic groups. Studies that report data on MVC fatalities of pedestrian workers focus on construction and law enforcement workers, both of which are occupations that would be expected to spend more time as pedestrians compared to other workers [29,37,146].

The practice of excluding pedestrian workers from MVC research ignores a population at disproportionately high risk, as pedestrians are at high risk of serious injury or death even when struck at low vehicle speeds. For a pedestrian, the average risk of death is 10% at an impact speed of 23 mph, rising to 75% at 50 mph and 90% at 58 mph. At 46 mph, the risk of serious injury is 90% [65]. Including pedestrian workers in research opens up opportunities to develop injury prevention strategies tailored to their needs and may benefit communities of color, who are disproportionately employed in jobs where they are often pedestrians.

As mentioned previously, persons who belong to communities of color are more likely than White persons to rely on public transit for commuting and at-work transportation, thereby increasing their exposure to traffic as pedestrians [147]. Reliance on public transit may be tied to a lack of access to a vehicle, which can close off some employment opportunities (job types and job locations) and lead to longer commutes.

#### 4.2.6. Increase Geographic Scope of Analyses of Work-Related MVCs

It must be acknowledged that the literature reviewed in this commentary is dominated by studies done in high-income countries, primarily the U.S., thus the disparities identified here reflect conditions in these locations. The disparities identified by U.S. research reflect the risks and benefits of a car-dominated society. U.S. research results cannot necessarily be applied to other high-income countries where trains, other forms of public transport, and two-wheeled vehicles are more widely used for commuting and work travel. They are also not applicable to low- and middle-income countries, where modes of transport, road infrastructure, working and social conditions, and healthcare differ from high-income countries. Additional research in low- and middle-income countries is needed to identify risk disparities in work- and commuting-related MVCs in these locations and strategies to remedy them.

For the U.S., FARS contains data on mile point, longitude, and latitude, creating the opportunity to overlay locations of fatal MVCs into geographic information systems [148]. Analyses of equity found in the transportation literature often examine data at the lowest possible level of geography available, such as census tract or block group level. In contrast, most of the published papers on work-related MVCs and injuries report data at the national level only, which lacks the granularity necessary to pinpoint the populations or geographic areas with the greatest risk disparities and identify the reasons for those disparities.

## 5. Conclusions

### 5.1. Need for a Multi-Level, Multi-Sectoral Approach

Advancing our understanding of disparities in work-related MVCs means moving beyond the examination of individual risk factors and addressing the broader social and institutional contexts in which MVCs occur. This commentary described how work-related MVC disparities are often related to what kinds of work people do, the type of employment arrangements they have, the way employers implement OSH policies, how employees commute to and from work, and where employees live. In applying the SEM to MVCs involving workers, most of the promising points of intervention would be at the organizational, community, and public policy levels. Interventions at a single level of the SEM are not likely to remedy health inequities for low-income workers [149]. Actions across sectors are critical: “Eliminating health inequities will benefit from a deliberate focus on health equity by public health agencies working with other sectors that impact health outcomes” [150].

### 5.2. Need to Improve and Expand Surveillance and Research to Understand Disparities and Mechanisms Leading to Inequities

Developing strategies to remedy racial, ethnic, and other disparities for work-related MVCs will require expanded surveillance and equity-centered research. Where possible, research should include mixed methods and incorporate variables at all levels of the SEM, to account for the likely multi-level sources of disparities and inequities. Other important steps include expanded surveillance and research for commuting-related crashes and pedestrian workers.

## Figures and Tables

**Figure 1 ijerph-20-06909-f001:**
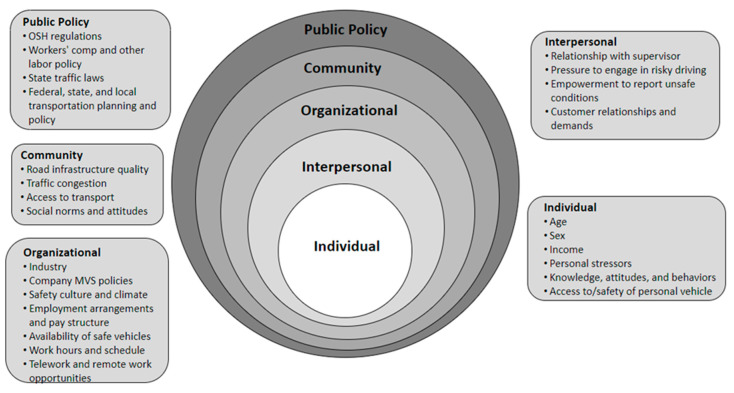
The social–ecological model applied to the risk of work-related motor vehicle crashes and injury. (Adapted from Ref. [53]).

**Table 1 ijerph-20-06909-t001:** Median annual incomes for driving- and traffic-intensive occupations by race and ethnicity: Full-time, year-round workers, United States, 2018.

Occupation	White, Non- Hispanic Workers		Black, Non- Hispanic Workers		Hispanic Workers (Any Race)	
	% Workforce	Median Income (USD)	% Workforce	Median Income (USD)	% Workforce	Median Income (USD)
All occupations	62.5	52,625	11.4	39,170	17.3	36,011
Security guards and gambling surveillance officers	43.4	35,629	30.8	31,249	18.8	30,705
Landscaping and groundskeeping workers	37.2	31,065	8.0	25,761	51.3	26,080
Couriers and messengers	52.9	40,021	19.5	37,013	21.2	35,369
Postal service mail carriers	59.6	57,915	18.9	51,801	13.1	50,918
Construction laborers	42.6	40,404	6.5	34,992	47.1	31,821
Highway maintenance workers	75.5	41,210	N/A	N/A	14.2	38,895
Driver-sales workers and truck drivers	56.2	48,188	16.7	44,776	22.5	40,587
Bus drivers, school	52.0	29,965	30.3	27,614	14.9	33,422
Bus drivers, transit, and intercity	33.0	36,480	39.8	45,228	19.6	37,477
Shuttle drivers and chauffeurs	36.3	30,832	25.3	32,176	24.4	31,243
Taxi drivers	29.1	29,593	26.4	30,565	23.7	30,539
Parking attendants	26.8	24,147	28.5	28,604	36.2	26,007
Refuse and recyclable materials collectors	44.2	41,803	19.3	32,013	32.2	31,682

Source: U.S. Census Bureau, 2021. Table 2. Full-time, year-round workers, and median earnings in past 12 months by race and Hispanic origin and detailed occupation, ACS 2018 [50].

## Data Availability

No new data were created or analyzed in this study. Data sharing is not applicable to this article.

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
