# Peer review of "Applying a Health Equity Lens to Work-Related Motor Vehicle Safety in the United States"

_ijerph, 2023, doi:10.3390/ijerph20206909_

Round 1
Reviewer 1 Report
This is a very comprehensive review of the disparity issues facing transportation. The authors are both seasoned researchers in occupational safety and health. I have not read a more thorough review of the evidence than this.
1) Line 51, p. 2 Add to "from unintentional injuries "Among all ages". I presume this is what you mean when you quote overdose and falls. but I believe the leading cause of death from unintentional injuries among adolescents and youth is still MV. (Check with WISQARS).
2. Page 4, can you also address SD affecting access to public transportation as another disparity?
3. Page 5 Flynn should be listed as (50), not 2022.
4. P 5 line 176 Same for McLeroy (1988)
5. Page 5. Cite source for Fig 1, or "adapted from..."
6. P 7, line 278. Mention here, all the other categories ...since you only mention 3 but enemerate about 4 or 5 (e.g. 3.3.1, - 3.3.4)
7. P. 11 and 12, lines 502-528. ( 3.5.5 and 3.5.6) distinguish more clearly the eifference between those two....sounds like they can be combined into one.
8. p 12, line 542-3 Define acronyms here if you haven't earlier.
why not add CDC-WISQARS and NEISS to the data linkage efforts, and also NHTSA/ MUCC data collection system in police reports.
9. p 13 can you add the need to also collect MV economic burden data such as CDC Injury Center's state cost models?
10. p 15 line 692 have you defined SEM earlier?
11. p 15, line 699 use the number of the reference, not the name and date.
12. Line 16, line 740. Define or give examples of a "pedestrian worker". Are they road crews, or police and/or occupants at a vehicle stop or road repair persons?
13. REFERENCES: SOME DATES ARE BOLDED, SOME ARE OT. sOME BOOKS AND REPORTS ARE ITALICIZED AND OTHERS ARE NOT (E.G. REF 56, 101, 114 and Ref # 47,62,91,107, 111).
Overall, an important review of what is known and what needs to occur to reduce disparities in transportation safety programs, surveillance, and research. WIll be a lasting contribution to the field.
Reviewer 2 Report
In this paper, the authors summarize the literatures on risk disparities for work-related MVCs by sociodemographic and employment characteristics and identify worker populations that have been largely excluded from previous research. The social-ecological model is used as a frame work to identify potential causes of disparities at five levels–Individual, Interpersonal, Organizational, Community, and Public Policy. Based on these results, meaningful research directions are clearly indicated.
This is a very interesting commentary paper suitable for publication in IJERPH. However, considering making it better, the following suggestions are made for the author's consideration.
1. Under section 2 there is only section 2.1. It is suggested that the primary heading of section 2 be replaced with "Disparities in work-related motor vehicle crashes", followed by secondary headings on factors such as gender and age. A similar situation also occurs in section 3.4.
2. The title of the section 3 is suggested to be changed, and it should be further clarified in the title what is to be accomplished by applying the social-ecological model.
3. The content of Section 5 (Conclusion) is more like a continuation or summary of the content of Section 4. It is recommended that the authors summarize the overall content of the paper from a more comprehensive perspective.
Reviewer 3 Report
Comments in attached document.
